# Use of Cyanobacterium Spirulina (*Arthrospira platensis*) in Buffalo Feeding: Effect on Mozzarella Cheese Quality

**DOI:** 10.3390/foods12224095

**Published:** 2023-11-11

**Authors:** Claudia Lambiase, Ada Braghieri, Carmela Maria Assunta Barone, Antonio Di Francia, Corrado Pacelli, Francesco Serrapica, Jose Manuel Lorenzo, Giuseppe De Rosa

**Affiliations:** 1Dipartimento di Agraria, Università degli Studi di Napoli Federico II, 80055 Portici, Italy; claudia.lambiase@unina.it (C.L.); cmbarone@unina.it (C.M.A.B.); antonio.difrancia@unina.it (A.D.F.); giderosa@unina.it (G.D.R.); 2Scuola di Scienze Agrarie, Forestali, Alimentari ed Ambientali, Università degli Studi della Basilicata, 85100 Potenza, Italy; ada.braghieri@unibas.it (A.B.); corrado.pacelli@unibas.it (C.P.); 3Centro Tecnológico de la Carne de Galicia, San Cibrao das Viñas, 32900 Ourense, Spain; jmlorenzo@ceteca.net; 4Área de Tecnología de los Alimentos, Facultad de Ciencias de Ourense, Universidade de Vigo, 32004 Ourense, Spain

**Keywords:** *Arthrospira platensis*, dairy buffaloes, PDO mozzarella cheese, sensory properties, consumer acceptability, willingness to pay

## Abstract

The high demand for PDO buffalo mozzarella cheese is leading to the use of new strategies for feeding supplementation. Spirulina is acknowledged as a valuable source of protein with antioxidant and immune-modulatory effects in humans and animals. This investigation aimed to examine the effect of Spirulina integration in buffalo diets on mozzarella cheese quality, sensory profile, consumer acceptability, and willingness to pay (WTP). The trial was carried out on two groups of 12 buffaloes that differed in Spirulina integration: 50 g/head/d before calving (1 month) and 100 g/head/d after calving (2 months). Both the bulk milk and mozzarella cheese samples from the two groups did not differ in chemical composition. However, Spirulina inclusion influenced the sensory quality of mozzarella cheese, which resulted it being externally brighter, with a higher butter odour and whey flavour and greater sweetness, bitterness, juiciness, tenderness, oiliness, and buttermilk release than the control. The consumer test showed that information about Spirulina affected consumer liking, causing them to be in favour of the Spirulina group, leading to a higher price for it. In conclusion, Spirulina inclusion in buffalo diets affected the sensory quality of mozzarella cheese. The provision of product information to consumers can be a crucial factor in determining their liking and WTP.

## 1. Introduction

In recent decades, buffalo breeding and milk production in Italy has been increasing, while in most other European countries, the number of buffaloes has been decreasing due to their use as draught animals, which has been replaced by mechanisation [1,2]. The growing progress of the Italian buffalo population, the nutritional and reproductive techniques, and the high quality of milk and meat products and markets have reached the top level in Europe. Many of the appreciated buffalo products come from milk, such as ricotta, provola, scamorza, and other cheeses, and from meat, such as steaks, roasts, ham, bresaola, and salami, but mozzarella cheese is the most important buffalo product on the Italian and international market (82% of consumption in Italy, and 18% for export, especially in Germany, France, UK, and USA) [1].

Almost all of the buffalo milk produced in Italy (mainly in the southern regions) is transformed into mozzarella cheese, a product of some specific areas of the Campania and Lazio regions that have been endowed with the Protected Designation of Origin (PDO) (Commission Regulation EC 103/2008) [3,4]. Buffalo mozzarella is a “pasta filata cheese”, which is sold very fresh (a few hours after the cheese is made) and is characterised by a soft, moist, and elastic texture, a bright and white porcelain colour, and a fresh flavour. Its sensory profile is influenced by the starter used in the cheesemaking process [5], the technological process [6], and most importantly, the milk composition, which is closely linked to animal feeding [7]. Although many authors have carried out studies on how a different diet can modify the chemical composition, sensory profile, or aromatic compounds of buffalo mozzarella cheese [8,9,10,11], none of them have used a “non-conventional” feed supplement. Recently, as reported in some reviews [12,13,14,15], there has been an increased focus on the impact of microalgae supplementation on the quality and production of milk in ruminants, as well as on the quality of eggs and chicken meat.

Microalgae are prokaryotic and eukaryotic organisms with a rapid growth rate, and they have the ability to grow in challenging environments, such as non-arable or marginal lands, and under different environmental conditions. They can be grown in either open or closed systems, under autotrophic or heterotrophic conditions, and their chemical compositions can be manipulated by changing the growth medium and/or the growth conditions. The species of microalgae with high concentrations of protein can partially replace conventional protein-based feeds, while those with high carbohydrate or lipid concentrations can be used as energy sources [13]. To date, *Arthrospira maxima* and *Arthrospira platensis*, both commonly known as Spirulina, are two of the microalgae registered under EU Regulation 767/2009 as feeding ingredients or animal feed [15]. Spirulina is considered one of the most nutritious foods due to its high concentrations of high-quality proteins (55–70% in dry weight, i.e., containing all essential amino acids) and vitamins and minerals such as B vitamins, iron, potassium, and many others. Currently, the effects of Spirulina as a feed supplement on animal health and/or production have been studied in different species, such as swine, poultry, and ruminants [12]. Although most ruminants have been considered in many studies [14,15], little is currently known regarding the performance and production of buffaloes when microalgae are added to their diets [16].

The objective of this study was to examine the impact of adding freeze-dried *Arthrospira platensis* biomass to the diets of dairy buffaloes on the chemical composition and sensory attributes of PDO mozzarella cheese. Furthermore, we assessed the influence of information regarding Spirulina’s health advantages on consumer acceptability and willingness to pay.

## 2. Materials and Methods

### 2.1. Animals, Diets, and Mozzarella Cheese Production

All cows enrolled in this study were handled in compliance with the Italian legislation concerning the protection of animals used for scientific purposes (DL n. 26, 4 March 2014), and all procedures involving animals used in this study were reviewed and approved by the Ethical Animal Care and Use Committee of the University of Napoli Federico II (Prot. 2020/007143 of 8/1/2020).

The trial was conducted on a buffalo dairy farm in the Southern Italian region of Campania (40°48′ N 15°01′ E; 14 m a.s.l.) between March 2021 and July 2021. Twenty-four late pregnant dairy buffaloes (*Bubalus bubalis*) were blocked for expected calving date and assigned to either a control or experimental group (hereafter referred to as C and S, respectively) balanced for cows’ number (*n* = 12), parity (3 secondiparous and 9 pluriparous), and milk yield from the previous lactation (2642 ± 503 kg/head and 2729 ± 492 kg/head for C and S, respectively).

The two groups were housed in 2 adjacent free-stall barns (13 × 10 m each) equipped with feed bunks, water troughs, and an outdoor exercise area. The experimental period lasted 120 days in total, including an adaptation period of 30 days before calving and a sampling period covering the early 90 days of lactation. Across the adaptation and sampling periods, all cows were fed the same standard total mixed ration (TMR) formulated to meet or exceed the recommended nutritional requirements for dairy buffaloes in the near dry period and early lactation, as recommended by Bartocci et al. [17] and Campanile et al. [18], respectively. As for the farm routine, both TMRs were offered once a day at 08:00 h, allowing for approximately 10% feed refusal. In the Spirulina group, the freeze-dried Spirulina was top-dressed just after the TMR delivery at a daily dose of 50 g/head until calving and 100 g/head during lactation. Appendix A lists the detailed ingredients and chemical compositions of the experimental TMRs (dry and lactation diets). The used Spirulina biomass (Table 1) was provided by ATI Bio-tech s.r.l. (Castel Baronia, Avellino, Italy). Cultivation was performed through industrial-scale outdoor open raceway ponds by using standard Zarrouk’s culture medium (20 °C; pH 10.25).

From 3 (±1) days of lactation onwards, buffaloes were milked two times a day (05:00 and 17:00 h) in the auto-tandem milking parlour equipped with a pipeline milking system that transports milk to a refrigerated (4 °C) tank.

After the first 30 days of milking, four mozzarella cheese-making sessions between the 2nd and 3rd month of lactation, with 2 sessions per month, were carried out by using refrigerated (4 °C) bulk tank milk that was separately collected at the evening milking session from each group (approximately 60 kg/group per milking). Bulk milk from each group was processed at the dairy farm, in separate vats, following the traditional procedure adopted for the PDO mark (Protected Designation of Origin) detailed by Sacchi et al. [3]. In brief, the raw milk was heated to 37 °C, and natural starter cultures and liquid calf rennet were added. The curd was then broken into small particles (2–3 cm) and left under whey until the pH reached 4.85. At this pH, the curd was manually stretched in water kept at 90–95 °C. Finally, 50 g of mozzarella (spherical shape) were cut mechanically, cooled in water, and left in brine (10% NaCl).

### 2.2. Measurements, Sampling Procedure, and Analytical Methods

#### 2.2.1. Feeds and Diets

Feed intake was monitored weekly on a group basis along the lactation period by subtracting feed refusals from the TMR offered. On the same days, individual feed and TMR samples were collected, dried in a forced air oven at 65° until a constant weight was reached, and ground to pass through a 1 mm screen to be analysed for the proximate composition according to the standard procedures described elsewhere [19,20].

#### 2.2.2. Milk and Mozzarella Chemical Compositions

Triplicate bulk milk samples (150 mL/group) were collected immediately after milking before each cheese-making session and analysed on the same days of collection in duplicate for protein, fat, and lactose using mid-infrared method (MilkoScan Minor Type 78100, Foss Electric, Hillerød, Denmark). Three mozzarella cheese samples were taken from each production batch the day after manufacturing. Samples were cut in slices and then minced by using a blender (LB20ES, Waring Commercial Blender, New Hartford, CT, USA), and they were separately analysed in duplicate (100 g per replication) for moisture, protein, fat, lactose, and salt contents using FoodScan Lab Analyzer (Foss Electric A/S, Hillerød, Denmark) operating between 2 scanning frequencies (850 and 1048 nm).

#### 2.2.3. Fatty Acids Composition

Fatty acid (FA) profiles of mozzarella cheese (10 g) were performed according to Vargas-Ramella et al. [21]. Fatty acids were transesterified and detected using a gas chromatograph (GC-Agilent 7890B, Agilent Technologies, Santa Clara, CA, USA) equipped with a flame ionisation detector (FID), a PAL RTC-120 autosampler (maintained at 250 °C and 64.2 mL/min of total flow rate), and a DB-23 fused silica capillary column (60 m, 0.25 mm i.d., 0.25 μm film thickness; Agilent Technologies) for the separation of Fatty Acid Methyl Esters (FAMEs). As standard, we used Supelco 37 Component FAME Mix (Sigma-Aldrich, St. Louis, MO, USA). Results were expressed as a percentage of total methylated FA. Values for individual FA < 0.1 were not reported. Atherogenic index (AI), thrombogenic index (TI), and hypocholesterolemic/hypercholesterolemic ratio (h/H) were calculated according to Chen and Liu [22].

#### 2.2.4. Amino Acid Determination

Amino acid (AA) profiles were determined according to the procedure described by Vargas-Ramella et al. [23]. The HPLC system used was an Alliance 2695 model equipped with a 2475 scanning fluorescence detector (Waters, Milford, MA, USA). To control the system operation and results management, an Empower 2^TM^ advanced software (Version 1.04.1037; Waters, Milford, MA, USA) was used. Separations were carried out using a Waters AccQ-Tag column (3.9 × 150 mm, with a 4 mm particle size) with a flow rate of 1.0 mL min−1 at 37 °C. Detection was carried out via fluorescence with excitation at 250 nm and emission at 395 nm. The AAs were quantified using the external standard method using an AA standard (Amino Acid Standard H, Thermo, Rockford, IL, USA) and identified via the retention time. The amount of each AA that was detected was expressed as mg/100 g sample.

#### 2.2.5. Minerals Determination

Initially, a 3 g sample was subjected to incineration at a temperature of 600 °C using a muffle furnace (Car-bolite RWF 1200, Hope Valley, UK). Afterwards, 10 mL of 1 M HNO_3_ was added, and the sample was further incinerated at 600 °C. Standards were diluted and used to calibrate the IPC-MS for mineral analysis. The incinerated samples were analysed for mineral contents (Ca, Fe, K, Mg, Mn, Na, P, Zn, and Cu) by using inductively coupled plasma mass spectrometry (ICP-MS) (Thermo electron X7 inductively coupled plasma mass spectrometry, Model X series, Oxford, UK). For the calibration of the IPC-MS for mineral analysis, standards were diluted and used. The ICP-MS operating conditions were as follows: nebuliser gas flow, 0.91 L/min; radio frequency (RF), 1200 W; lens voltage, 1.6 V; cold gas, 13.0 L/min; auxiliary gas, 0.70 L/min [24]. Results were expressed as mg/100 g of sample.

#### 2.2.6. Volatile Organic Compound Determination

The volatile organic compound (VOC) extraction was carried out using a solid phase microextraction (SPME) device (Supelco, Bellefonte, PA, USA), including a 10 μm length fused silica fibre coated with a layer (50/30 μm thickness) of divinylbenzene/carboxen/polydimethylsiloxane (DVB/CAR/PDMS), following the procedure described by Echegaray et al. [25]. Briefly, 1 g of minced mozzarella cheese was weighted in a 24 mL glass vial closed with a screw cap equipped with a laminated Teflon–rubber disc. The samples were kept at 35 °C for 15 min, allowing for the equilibration of VOC in the headspace. Then, the fibre, already conditioned at 270 °C for 60 min via heating in a gas chromatograph injection part, was inserted in the headspace of the vials through the septum and exposed to the headspace for 30 min at 35 °C. The splitless mode was used to inject the samples. Helium was used as carrier gas, with a linear velocity of 40 cm s^−1^. The total run time of 49 min and 30 s was divided as follows: 10 min at 40 °C (isothermal), and then increased to 20 min at 0 °C (5 °C/min), and finally, 250 °C (20 °C/min), maintained for 5 min. Both the injector and detector temperatures were set at 260 °C. The gas chromatograph used for the VOC detection was 6890 N (Agilent Technologies, Santa Clara, CA, USA) equipped with a DB-624 capillary column (30 m, 0.25 mm i.d., 1.4 μm film thickness; J&W Scientific, Folsom, CA, USA) and coupled to a mass selective detector, 5973 N (Agilent Technologies). VOCs (expressed as area units (AUs) × 10^4^ per g of product) were identified by comparing the detected mass spectra with those reported in the NIST05 library (National Institute of Standards and Technology, Gaithersburg, MD, USA) (matching was considered when coincidence was greater than 80%), and/or by comparing the mass spectra and retention times with the authentic standards (Supelco, Bellefonte, PA, USA), and/or by calculating the retention index against a set of standard alkanes (C5-C14) (for calculating Kovats indices, Supelco 44585-U, Bellefonte, PA, USA) and by comparing them with the data reported in the literature.

#### 2.2.7. Colour Evaluation

The colour analysis of mozzarella cheese was performed by implementing a Computer Vision System (CVS), following the method by Girolami et al. [26,27], on the same batch of samples used for the sensory analysis (QDA) before tasting and for each cheese-making session. Briefly, each mozzarella slice was cut in two; one half was used in order to evaluate the inner colour, and the other half was used to evaluate the outer colour of the sample. The images were captured with a digital camera (CANON EOS 450D, 12.2 Megapixel; Canon Inc., Tokyo, Japan) at a distance of 30 cm, in a black wooden box, under four fluorescent lamps at 50 cm from the sample and angled at 45° to ensure uniform light intensity on the sample. The camera was connected to an NEC MultySync LCD monitor with sRGB colour space (standard RGB) and a resolution of 1600 × 1200 pixels. The monitor was calibrated, and the ICC monitor profile was created using the Eye-One Match 3.2 software. The rendering intent used was of perceptive type. Colour evaluation was performed using Adobe Photoshop CS3 software. RGB images were transformed into L*, a*, and b* values after acquisition via RAW photographs. Particularly, three areas of the digital image of each sample were spotted by the cursor, and the colorimetric characteristics were measured.

### 2.3. Sensory Analyses

Sensory analyses included sensory profile, assessed via quantitative descriptive analysis (QDA) according to Murray et al. [28], consumer liking, and willingness to pay (WTP).

#### 2.3.1. Quantitative Descriptive Analysis (QDA)

To perform QDA, fifteen regular eaters of buffalo mozzarella cheese (i.e., consuming it at least once a week) were enrolled. Among them, ten panellists (5 males and 5 females between 24 and 32 years of age) were chosen according to their capacity to identify the 4 basic tastes [29], as recommended by ISO recommendations (ISO 8586, [30]). A labelled scale with intervals of intensity was used to train panellists on the use of the 100 mm unstructured intensity linear scale (0 = absent; 100 = very high) provided for QDA [31]. In a subsequent session, the assessors tasted the mozzarella cheese samples and, based on specific works in the literature [11,32], they developed and agreed on a vocabulary of attributes (about appearance, odour, taste, flavour, and texture) and their definitions (Table 2). Their subsequent training was performed in individual sensory booths (ISO 8589) [33] by using a specific frame of reference [34] to identify the intensity ranges for low and high intensity for each attribute. Then, panellists tasted the samples again with the two preferences of intensity for each attribute in blind conditions. This training phase was essential to calibrate the performance of the panellists in terms of repeatability, discrimination, and agreement. For the test, two 50 g mozzarella cheese samples for C and S were administered in a randomised order at a temperature of 13 °C. Each sample was identified by a three-digit number code. To avoid the effect of appearance on the perception of odour/flavour, taste, and texture attributes, the assessors evaluated the intensities of these attributes of the first cheese under red light, while the intensities of the appearance parameters were assessed on the second cheese under white, fluorescent lighting. Each product was evaluated in 3 replications. The interval between consecutive samples was roughly 10 min for each product; assessors were recommended to drink a sip of water and to eat a piece of smith apple between two consecutive tastings to reset the effect of the previous sample. The Smart Sensory box 2.3.5 platform was used to manage the QDA sessions (Smart Sensory Solution, Sassari, Italy).

#### 2.3.2. Consumer Test and Willingness to Pay

A total of 68 consumers (42 females and 26 males, from 18 to 60 years old) participated in the test. The reduced number of participants in the consumer trial (68 versus 100) was a consequence of the COVID-19 pandemic, which posed a challenge to recruitment. Consumers among regular eaters of buffalo mozzarella cheese (i.e., consuming this product at least once a week) and who were available over the study period were recruited by phone. The study was conducted in agreement with the guidelines of the Declaration of Helsinki and the Italian ethical requirements on research activities and personal data protection (D.L. 30.6.03 n. 196).

Firstly, they were asked to taste 50 g of mozzarella cheese from both groups (C and S) and rate their liking in blind conditions, that is, they did not receive information on the products (perceived liking, B). For each sample, consumers rated their overall liking and their liking for the appearance, taste/flavour, and texture using a 9-point hedonic scale, where the 9 categories ranged from “dislike extremely” to “like extremely”, with a central point (5) corresponding to “neither liked or disliked” [35,36].

On the same day, in the second test, the subjects received two sheets with information concerning the nutritional quality of mozzarella cheese, and they were asked to carefully read the information and give their liking expectation for that product without tasting the samples (expected liking, E). In detail, the information about the two products was as follows:Information for sample 1—Mozzarella di bufala Campana produced with milk from conventional breeding. “Mozzarella di bufala Campana” is produced exclusively with fresh whole buffalo milk according to the specific disciplinary published in the Italian Official Gazette n. 258 of 6.11.2003. Mozzarella di bufala Campana is porcelain white in colour, has a characteristic and delicate flavour, a fat content of at least 52% (on the dry matter), and a maximum moisture content of 65%.Information for sample 2—Buffalo mozzarella produced with milk from conventional breeding and Spirulina dietary supplement. “Mozzarella di Bufala Campana” is produced exclusively with fresh whole buffalo milk according to the specific disciplinary published in the Italian Official Gazzette n. 258 of 6.11.2003. Mozzarella di bufala campana is porcelain white in colour, has a characteristic and delicate flavour, a fat content of at least 52% (on the dry matter), and a maximum moisture content of 65%. Dietary supplementation with Spirulina (a natural product arising from the freeze-drying of microalgae) makes it possible to obtain a mozzarella with antioxidant, anti-cholesterol, and immune-stimulating properties.

The third test was performed the day after. Consumers received both products (C and S), each accompanied by the corresponding information. They had to read the information before tasting the sample and reporting their liking scores (actual liking, A). For the QDA, tests were performed in a controlled sensory analysis laboratory [37] equipped with individual booths under white fluorescent lighting. Consumers were asked to clean their mouths with some water and to eat a piece of smith apple between each sample evaluation to try to make the palate conditions similar for each sample. Samples, identified by a 3-digit number code, were served in random order. The Smart Sensory box 2.3.5 platform was used to manage the consumer test sessions (Smart Sensory Solution, Sassari, Italy).

#### 2.3.3. Vickrey Auction

After the third test, two 250 g packs of mozzarella (C and S) were auctioned off. A second-price Vickrey auction [38] was performed to assess consumers’ WTP mozzarella cheese produced from buffalo milk fed with Spirulina supplementation, according to Napolitano et al. [39]. A short explanation of the procedure for the auction was given to the participants, and it was clarified that the proposed offers involved a commitment to purchase the product. Participants signed a consent agreement to the procedure and received EUR 10 (in cash). Then, they were formally trained on the use of Vickrey’s second-price auction. Each consumer separately had to write, on a paper, the maximum price to pay for one unit of product evaluated. The winner (i.e., the consumer submitting the highest value) had to purchase the product not at the proposed price, but at the second highest offer submitted. Thus, one of the participants was allowed to purchase a product at an equal or lower price than what they would normally have agreed to pay. In the case of more consumers offering the same highest offer, only one of them, randomly chosen by another consumer, would be selected as the winner. It was also clarified that the aim of the auction was to understand the value that the product had for consumers, and not its commercial value. To verify that all participants correctly understood the procedure, some practice with snacks was carried out. At this point, the test was performed. To prevent the participants from becoming less motivated in winning more products, it was explained that the mozzarella cheese pack would be evaluated under different information conditions, and only one condition, randomly chosen by a consumer, would be used for the actual auction.

### 2.4. Statistical Analyses

All the data collected were subjected to a Kolmogorov–Smirnov normality test. The chemical composition (i.e., fat, protein, lactose, fatty acid profile, amino acid profile; minerals for both milk and mozzarella; and moisture, salt, VOC, and colour for mozzarella only) was analysed via one-way analysis of variance (ANOVA) (GLM, general linear model), with group as a factor.

To verify panel performance, sensory data from the QDA test were subjected to a “Fixed” ANOVA model with 2 levels as products, 10 levels as assessors, and 3 levels as replications and their interaction (products x assessors, products x replications, and assessors x replications). To evaluate panel performance, we analysed the significance of replication effects and the assessors x products and assessors x replications interactions. The analysis of the product effect enabled us to evaluate the significant differences between the products in terms of the perceived intensity for each attribute listed in Table 2, and the perceived attributes’ intensity differences. Consumers’ liking data were analysed via one-way ANOVA (GLM) to identify the most liked product. Student’s paired *t*-test was used to evaluate differences between mean scores obtained for the two products and for the same products under different conditions (tasting only, information only, and tasting with information). Data collected for the validation of information were subjected to analysis of variance with type of information as a factor. All the statistical analyses were performed by using SAS software (2009).

## 3. Results and Discussion

### 3.1. Spirulina Effects on Mozzarella Cheese’s Chemical Composition

The milk composition was affected by Spirulina supplementation only in terms of protein, lowering the percentage from 4.59 to 4.31 (SEM 0.07, *p* < 0.05), while the fat and lactose contents were not influenced by Spirulina inclusion in the buffalo diets. The same trend was observed in the mozzarella cheese’s chemical composition; the fat, lactose, and moisture contents were not different among the two groups, while the protein amount was tendentially higher in the C group ((18.01 vs. 17.12 ± 0.33%; *p* < 0.10 (mean ± SEM)) than the S group. Our results do not agree with the increases in protein, fat, and lactose found by Šimkus et al. [40] as effects of Spirulina supplementation in cows, and the findings of Boeckaert et al. [41] and Stamey et al. [42], who registered an increase in milk fat. Despite the small differences in the protein content, the AA profiles of the mozzarella cheese samples did not change with Spirulina supplementation (Table 3). Among the EAAs, LEU and LYS are the most abundant in both products, tending to be higher in the S samples, which could be due to the AA profile of the added Spirulina. Microalgae in general are good sources of EAAs. The milk from the S group contained 27% more EAAs and 23% more NEAAs than the C group. The presence of EAAs in milk and dairy products, especially branched-chain amino acids, is important in terms of their benefits for human health [43]. Recently, Cacciola et al. [44,45] reported interesting results about a possible beneficial effect of delactosed buffalo milk whey by-product on colorectal human carcinogenesis. It is well known that primary proteolysis occurs during mozzarella processing [46], and the effect of the composition of milk protein fractions (relative contents of alpha S1 casein and beta casein) on milk coagulation properties has been widely studied [47,48].

Table 4 shows no significant differences in the FA profiles of the mozzarella cheese samples between the C and S groups, nor in the calculated nutritional index. Manzocchi et al. [49] did not observe any significant differences in the dairy cow milk FA content when substituting 5% of soybean meal with Spirulina in their diets, except for the total n-6 FA, which was higher in the control milk samples. This highlights that the diet supplemented with Spirulina lowered the apparent transfer efficiency of the total n-6 FA. In contrast, numerous authors have reported that replacing feed with Spirulina yields a significant improvement in the FA profile of ruminants’ milk. Christodoulou et al. [50] examined the effects of three levels of Spirulina substitution (i.e., 5%, 10%, and 15%) in the diets of ewes. Their findings revealed that only the highest level of substitution increased the PUFA and ω-3 PUFA contents, while a mere 5% of substitution was adequate to decrease the AI. Christaki et al. [51] added 40 g of powdered Spirulina to cow diets and observed a decrease in the SFA amount, while the MUFA and PUFA contents in the milk increased at the 45th day of experimentation. As stated by many authors, species and breeds strongly affect the animals’ responses to diets [14,15]. This might partially explain why the buffalo mozzarella cheese chemical compositions did not differ among the two groups in our study. Moreover, the amount of Spirulina included in the S diets for the buffaloes could be too low to see changes/improvements in the milk and cheese compositions. To date, most of the studies in which microalgae were tested for animal feeding showed inconsistent and contradictory effects, probably due to the high variability among the studies, the doses of microalgae and species, the experimental period, and the percentage of forage in the diet [14]. However, microalgae use in animal feeding originated with the objective of improving the nutritional quality, particularly in terms of the FA profile, of milk and dairy products [15].

The mozzarella cheese’s mineral content was not affected by the diets, as shown in Table 5. As it is well known, the major mineral that is present in buffalo milk, and thus in buffalo mozzarella cheese, is calcium, while the second most predominant mineral is phosphorus. Gulzar et al. [52] reported that the mineral content is influenced by the cheese process, with the ash content reducing proportionally as the milling pH decreases. They explained that acid development solubilises micellar calcium phosphate, leading to mineral expulsion in whey. Specifically, the calcium levels decrease in line with the milling pH reduction, although there is uncertainty about the influence of this cheese-making step on the potassium and sodium levels. The buffalo milk mineral content is subject to variation due to several factors, including the season, environment, diet, stage of lactation, animal breed, and genetics [53]. The Mediterranean buffalo has been reported to possess the highest magnesium content in milk, whilst the greatest amount of minerals found in buffalo milk occurred during the summer season [54]. In our study, it was found that the mineral content was consistent across the samples because both groups were homogeneous, consisting of the same breed, and were fed the same diet. Therefore, the inclusion of Spirulina powder did not affect the mineral percentage in the buffalo mozzarella cheese.

The results of the analysis on VOCs can be found in Table 6. In total, 94 compounds were identified, and the most prevalent class in both groups is “others”. This group covers all of the individual VOCs that do not belong to the main classes considered in this study. Although the less abundant classes are the same in both samples, including esters, aldehydes, halogenated hydrocarbons, sulphur, and nitrogen compounds (from highest to lowest), the most abundant VOC classes are different between the two groups. In the C samples, the most and least frequent classes are alcohols, ketones, hydrocarbons, and aromatic hydrocarbons, in that order. In contrast, in the S samples, the most frequent classes are hydrocarbons, alcohols, aromatic hydrocarbons, and ketones, in that order. The only class that significantly differs from the others is aromatic hydrocarbons, which are higher in the C samples than the S samples, despite being ranked fifth on the list of VOC classes for the C group (in descending order) and fourth for the S group. However, the individual VOCs that were found to be significantly different (only 4% of the VOCs detected) do not belong to this class. Specifically, toluene (*p* < 0.05), isobutyl acetate (*p* < 0.05), and acetic acid 2-phenylethyl ester (*p* < 0.01) were found at higher levels in product C, while 1 Pentanol (*p* < 0.01) was detected at lower levels (*p* < 0.01) (Table 6). Unlike the study by Sabia et al. [10], where the most prevalent VOC class was ketones, particularly in the cheese from buffalos that were fed wrapped ryegrass silage, our study found that ketones ranked third and fifth in the VOC lists of the C and S groups, respectively. Moreover, the level of ketones was higher in the C samples compared to the S samples (though not significantly). This is similar to the study by Sacchi et al. [3], where ketones were more abundant in the control sample than the experimental ones. Ketones are common in many dairy products, and their origin can be attributed to SFA β-oxidation. Some studies suggest that these compounds are derived from animal feeds, and silage is the primary source of ketones [3]. Further, animals consuming silage-based diets may result in the production of alcohols [55]. There are numerous metabolic pathways implicated in the alcohol biosynthesis found in cheese. These pathways include the reduction of methyl ketones and aldehydes, the degradation of linoleic and linolenic acids, as well as AA and lactose metabolism. Table 6 shows that the alcohol class is higher in the C samples in comparison to the S samples, although not significantly different, and they are the same as the ketones levels. Our study involved two groups of animals that were fed the same diet, with the same amount of silage. Therefore, the variations in the levels of ketones and alcohols could be attributed to the addition of Spirulina, even in small quantities. Although there is no significant difference in the levels of halogenated hydrocarbons between the two groups, it should be noted that the level of this class is four times greater in the C samples than in the S samples. The mozzarella cheese production process impacts both the quantity and quality of the VOC composition. High temperatures during stretching can cause compound loss, while microflora activity during curd ripening can generate new compounds. Additionally, enzymes and milk bacteria are involved in the development of the cheese’s flavour. The aroma of cheese is also the result of the interaction of volatile and non-volatile chemical compounds, mainly concentrated in the water-soluble fraction, resulting from the transformation of the primary components (fat, proteins, and carbohydrates) and the action of bacteria present during milk processing. Therefore, understanding the production process is crucial to regulate the VOC composition of mozzarella cheese [3]. However, the flavour of mozzarella cheese is not determined by the quantity of a particular compound/VOC, but by the equilibrium of all of the VOCs present.

The results about the instrumental colour of mozzarella showed that Spirulina integration was responsible of the higher values of the L*, a*, and b* indices in both the external and internal surfaces (Table 7). It is likely that the blue phycocyanin, one of the two pigmented antioxidants found in Spirulina, may have contributed to an increased lightness with a blue tint. However, it is worth noting that Park et al. [56] did not find a significant correlation between the L* value, pigment content, or antioxidant activity of Spirulina. If this finding is confirmed in future studies, it would be a favourable outcome, considering the high demand for porcelain white mozzarella among consumers.

A correct instrumental measurement of colour is useful at the stage of analysing consumer preferences for the research, development, and improvement of cheese-making methods. When observed under a light source, PDO buffalo mozzarella cheese should have a porcelain white colour and a glowing appearance. The white colour is given by casein micelles, while the yellowish hues are imparted by carotenoids in green fodder, which are not absorbed in the cattle species and are therefore released in the milk. In contrast, buffaloes, sheep, and goats are able to assimilate and transform these compounds, so the colour of their milk tends to remain white.

### 3.2. Spirulina Effects on Mozzarella Cheese’s Sensory Quality

As for the QDA, no significant product x replication or product x assessor interactions were observed, suggesting that the training program and the reference frame used in this study were efficacious in ensuring the high reliability of the panel (i.e., products were not evaluated differently in different replications or by different assessors).

Table 8 shows a significant diet effect on the sensory profiles of mozzarella cheeses. The samples from the S group were evaluated as being brighter (*p* < 0.05) with a lower white inner colour (*p* < 0.01) than the C group. The S mozzarella cheeses were perceived with a higher butter odour (*p* < 0.01), a higher whey flavour, and a sweeter and more bitter taste than the samples from the C group (*p* < 0.05). Furthermore, the panellists perceived higher oiliness (*p* < 0.01) and moisture (*p* < 0.0001) intensities in the S samples with a greater milk release when cutting (*p* < 0.001) and a tendentially higher tenderness (*p* < 0.10) than in the mozzarella cheeses from the C group, which, on the contrary, were perceived as being more grainy, cohesive, screechy, and having a greater consistency when cutting than the S mozzarella cheeses (*p* < 0.01).

In a recent study on buffalo mozzarella cheese [11], changes in texture were attributed, at least in part, to the corresponding fatty acid composition, which was lower in saturated fatty acids, such as C14:0, and richer in unsaturated fatty acids, such as C18:1, in products obtained from animals that were fed fresh fodder. The lower melting point of unsaturated fatty acids can produce softer cheeses. Furthermore, some attributes (e.g., bitter taste, flakiness, and grainy texture) may be influenced by more intense proteolysis that occurs in the summer, which, in turn, may contribute to the softening of the mozzarella and may have a direct effect on the flavour through the production of short-chain peptides and amino acids [57].

The greater oiliness could also be due to a different acid composition in the fat. For example, in Friesian cattle, Christaki et al. [51], following the administration of 40 g/head/d of Spirulina, observed a reduction in the content of saturated fatty acids in milk, with a consequent increase in MUFA and PUFA compared to the control. The sensory characteristics of fresh mozzarella largely rely on the raw materials and production techniques used. Further investigation is required to evaluate the influence of diverse components, including VOCs, AAs, FAs, and peptides, on each sensory attribute.

The data on the distribution of individual preferences (Table 9) show that the information received is able to direct the consumers’ judgements.

The consumers in blind conditions (P) evaluated both samples from the C and S groups to be greater than the middle value (i.e., 5) (Table 10), meaning there were satisfactory sensory properties in both products. Moreover, the blind evaluation was not influenced by the animals’ diet, nor by the ages and sexes of the consumers. No difference was observed between the two groups for expected (E) liking. Likewise, the actual (A) liking was only influenced by diet, with a higher rate for the S group samples (7.29) than the C group samples (6.85). Table 10 shows that for both groups, the expected liking was significantly higher than the blind evaluation (*p* < 0.01), which means there was a negative disconfirmation: consumers evaluate the mozzarella cheese to be lower than their expectations. There was no difference between the actual and perceived liking of the C group samples, which means that the information does not increase the product evaluation; on the contrary, for the S mozzarella cheese, the information strongly influenced the actual liking (*p* < 0.001), as it was close to the expected acceptability. This means that there was a complete assimilation, because the information given to the consumers brought them to similarly evaluate the samples in both the actual and expected conditions (i.e., with and without tasting the mozzarella cheese).

Many aspects of the product can be used by consumers to make their food choices. Grunert et al. [58] identified four main dimensions of quality for dairy products: hedonism, health, convenience, and process. A few of them can be experienced before purchase (e.g., colour), while most of them can be perceived after purchase (e.g., sensory properties), or may never be perceived (e.g., healthiness and appearance ethical). The latter must be communicated to the consumer, as they are characteristics that cannot be perceived before nor after the purchase [59]. Therefore, regarding these characteristics, consumers are forced to develop expectations to guide their food choices. In particular, to develop expectations on product quality attributes, consumers can use both intrinsic (e.g., holes in the cheese, colour of the external rind, etc.) and extrinsic (e.g., price) characteristics. However, consumers tend to rely mostly on extrinsic characteristics provided to them in the form of product information [60]. In particular, ethical concerns, such as environmental pollution and animal welfare, are becoming increasingly important in the hierarchy of purchasing motivations for animal products. Blockhuis et al. [61] highlighted that animal welfare is increasingly recognised as an important component of quality for consumers of animal products. Numerous studies have been conducted on the effect of information on food liking [62,63]. For example, the effect of information relating to animal welfare on lamb [64] and beef [65] liking has been studied, as well as the effect of information relating to organic production on the acceptability of foods and beverages [66,67]. All of these studies have shown that information-induced expectations can influence the perception of quality. Therefore, if expectations receive a positive disconfirmation (when the satisfaction score of the product tasted without external information is higher than expected) or a negative one (when the product is worse than expected), the assimilation model is generally applicable. According to this model, when external information is provided, the hedonic tests highlight a shift in acceptability towards expectations and reach different values from those obtained by tasting the same food without external information [68]. In both groups, the expected liking was significantly higher than the liking expressed in the blind conditions (*p* < 0.01), thus indicating that a negative disconfirmation occurred: the consumers found the mozzarella to be less pleasant than expected. In these conditions, generally, actual liking moves in the direction of expected liking for group C; no significant difference was observed between the actual and perceived liking. Thus, in this case, there was no assimilation, as the information did not improve the acceptability of the product (Table 10). Conversely, regarding the mozzarella cheese obtained from the S group, the information greatly improved the actual liking (*p* < 0.001), which moved in the direction of the expectations. According to Blokhuis et al. [61], the perception of food quality is determined by the welfare of the animals producing that food together with the overall nature and safety of the final product. In particular, the assimilation in this case was complete, since no difference was observed between the actual liking (expressed by the consumers with both sensory stimuli and information available) and expected liking (expressed by the consumers with only the information available). The complete assimilation observed for the S product is probably due to the important role played by information in determining the actual liking of mozzarella with Spirulina. This information is able to respond to some of the main and most current consumer concerns, such as the nutraceutical properties of products and animal welfare.

Napolitano et al. [64] showed that consumers are influenced by animal welfare information and shift their WTP in the direction of their expectations. In particular, the discrepancy between the expected liking and the actual willingness to pay was not fully assimilated, indicating that it was also expressed in relation to other aspects (for example, the sensorial properties of the products). Experimental auctions are able to place consumers in real situations where they can show their true preferences. In particular, the second-price Vickrey’s auction has been widely used to assess consumers’ WTP for real goods, including food [38], and the values that consumers place on food safety [69] and animal welfare [39]. Under this specific type of auction, consumers are individually asked to submit a sealed bid corresponding to the highest price they would agree to pay for a particular product. The highest bidder (i.e., the winner), by paying the second highest price, has the opportunity to purchase a product at a price that is equal to or, more often, lower than the value they attribute to the product [70]. Currently, consumers are not looking for the cheapest food, but the best quality/price ratio, i.e., the maximum benefit for what they are willing to spend [71]. In our study, it was observed that gender had no influence on the overall offer, while it was found that the consumers offered EUR 4.05 (16.20 EUR/kg) for a 250 g pack of Spirulina mozzarella cheese compared to EUR 3.52 (14.08 EUR/kg) for the control mozzarella cheese (F1,128 = 8.73; *p* = 0.0074).

## 4. Conclusions

The inclusion of 100 g of freeze-dried Spirulina per day in the diets of buffaloes does not alter the chemical composition of mozzarella cheese or the characteristics identified using chromatographic instruments, including the fatty acid profile or the classes of volatile organic compounds. Nevertheless, this small amount of supplementation affects the sensory characteristics of mozzarella cheese, as assessed by the panel of experts. The experimental mozzarella cheese was found to be brighter, sweeter, more bitter, juicier, more tender, and oilier, with higher buttermilk release, butter odour, and whey flavour than traditional buffalo mozzarella cheese. In addition, providing information about the nutraceutical properties of Spirulina and its possible beneficial effect on animal health to consumers can have a strong effect on their liking, as they rated the experimental mozzarella greater than the conventional one, and expressed a higher willingness to pay for the mozzarella cheese produced by the buffalo fed with Spirulina. Further studies with a higher level of Spirulina are necessary in order to confirm these results.

## Figures and Tables

**Table 1 foods-12-04095-t001:** Chemical composition of freeze-dried Spirulina.

Proximate Composition	%	Fatty Acid Profile	%	Amino Acid Profile	mg/100 g	Minerals	mg/100 g
Moisture	8.29	SFA	40.11	His (E)	1108.26	Ca	83.01
Ash	8.78	C 16:0	38.33	Arg (E)	4105.04	Fe	25.76
Fat	5.84	C 18:0	1.06	Thr (E)	3094.66	K	1879.29
Protein	64.95	MUFA	10.07	Val (E)	4163.97	Mg	178.44
		C 16:1 n-7	7.28	Met (E)	975.55	Mn	2118.37
		C 18:1 n-9	1.41	Lys (E)	2819.49	Na	1442.96
		PUFA	49.82	Ile (E)	3701.69	P	933.98
		C 18:2 n-6	23.89	Leu (E)	5527.34	Zn	0.75
		C 18:3 n-6	23.50	Phe (E)	2918.31	Cu	0.28
		C 20:3 n-6	1.40	Asp	5675.80		
				Ser	3206.42		
				Glu	9685.14		
				Gly	3055.51		
				Ala	4447.83		
				Pro	2381.84		
				Cys	458.28		
				Tyr	2616.80		

SFA, Saturated Fatty Acid; C 16:0, Palmitic Acid; C 18:0, Stearic Acid; MUFA, Monounsaturated Fatty Acid; C 16:1 n-7, Palmitoleic Acid; C 18:1 n-9, Oleic Acid; PUFA, Polyunsaturated Fatty Acid; C 18:2 n-6, Linoleic Acid; C 18:3 n-6, γ-Linolenic Acid; C 20:3 n-6, Dihomo-γ-Linolenic Acid; (E): Essential Amino Acid; His, Histidine; Arg, Arginine; Thr, Threonine; Val, Valine; Met, Methionine; Lys, Lysine; Ile, Isoleucine; Leu, Leucine; Phe, Phenylalanine; Asp, Aspartic Acid; Ser, Serine; Glu, Glutamic acid; Gly, Glycine; Ala, Alanine; Pro, Proline; Cys, Cysteine; Tyr, Tyrosine; Ca, Calcium; Fe, Iron; K, Potassium; Mg, Magnesium; Mn, Manganese; Na, Sodium; P, Phosphorus; Zn, Zinc; Cu, Copper.

**Table 2 foods-12-04095-t002:** List of attributes used by the 10-member trained panel for buffalo mozzarella cheese sensory profiling.

Attribute	Definition
*Appearance*	
Colour uniformity	Overall colour uniformity
Surface uniformity	Product surface free of holes and granules
Inner colour	Intensity of the cheese’s inner colour (from white to ivory)
Brightness	Shininess or glossiness of the surface
Number of eyes	Average number of eyes in the cheese mass
Uniformity after cutting	Cut surface free of holes and granules
Buttermilk release	Amount of buttermilk released after cutting
*Odour*	
Milk	Odour arising from milk at room temperature
Butter	Odour arising from butter at room temperature
Yoghurt	Characteristic odour of plain whole yoghurt
*Taste*	
Saltiness	Fundamental taste associated with sodium chloride
Sweetness	Fundamental taste associated with sucrose
Bitterness	Fundamental taste associated with quinine
Sourness	Fundamental taste associated with citric acid
*Flavour*	
Fruity	Odour associated with fruits such as pineapple
Whey	Characteristic whey flavour
*Texture*	
Tenderness	Minimum force required to chew cheese sample; the lower the force, the greater the tenderness
Oiliness	Amount of oily/fatty feeling in the mouth during chewing
Moisture	Moisture released by the cheese in the mouth during early mastication
Graininess	Perception of particles (grains) in the mouth
Cohesiveness	Degree to which a cheese sample holds together or adheres itself after chewing
Screechiness	Friction of the product against the teeth
Shear consistency	Consistency of the sample during cutting

**Table 3 foods-12-04095-t003:** Amino acid profiles of mozzarella cheese from control (C) and Spirulina (S) groups.

Amino Acid (mg/100 g of Sample)		Group		SEM
*Abbreviation*	*Common Name*	C	S	
Total EAA		6031.10	7040.75	736.86
His	Histidine	388.00	448.76	46.68
Arg	Arginine	552.39	633.09	55.73
Thr	Threonine	520.98	608.85	60.38
Val	Valine	793.33	931.31	103.68
Met	Methionine	278.41	333.20	48.91
Lys	Lysine	1039.48	1226.28	134.79
Ile	Isoleucine	645.86	753.29	83.53
Leu	Leucine	1176.33	1373.75	155.64
Phe	Phenylalanine	636.33	732.23	79.97
Total NEAA		7150.25	8360.97	930.26
Asp	Aspartic acid	857.74	1015.54	114.24
Ser	Serine	742.36	873.18	94.15
Glu	Glutamic acid	2868.14	3361.24	389.88
Gly	Glycine	255.70	297.84	33.40
Ala	Alanine	364.21	424.53	43.46
Pro	Proline	1305.20	1520.22	171.65
Tyr	Tyrosine	733.81	844.68	85.86
Cys	Cysteine	23.09	23.74	6.92

SEM, Standard Error of the Mean; EAA, Essential Amino Acid; NEAA, Not Essential Amino Acid.

**Table 4 foods-12-04095-t004:** Fatty acid (FA) profiles (% on total FAs) and nutritional index of mozzarella cheese from control (C) and Spirulina (S) groups.

Fatty Acids		Group		SEM
*Abbreviation*	*Common Name*	C	S	
C4:0	Butiric acid	2.74	2.76	0.034
C6:0	Caproic acid	1.77	1.79	0.024
C8:0	Caprylic acid	1.05	1.05	0.016
C10:0	Capric acid	2.09	2.10	0.029
C12:0	Lauric acid	2.60	2.61	0.035
C14:0	Myristic acid	11.50	11.63	0.138
C15:0	Pentadecylic acid	1.07	1.04	0.019
C16:0	Palmitic acid	35.02	35.31	0.397
C16:1 n-7	Palmitoleic acid	1.72	1.65	0.062
C18:0	Stearic acid	12.76	12.59	0.169
C18:1 trans 9	Elaidic acid	0.63	0.64	0.008
C18:1 trans 11	Transvaccenic trans-11-Octadecenoic acid	1.62	1.59	0.037
C18:1 n-9	Oleic acid	19.77	19.61	0.236
C18:2 n-6	Linoleic acid	1.85	1.90	0.058
C18:3 n-6	γ-Linolenic acid	0.22	0.23	0.014
C18:2 c9-t11 CLA	Linolenic acid isomer	0.66	0.64	0.015
C20:0	Eicosanoic acid	0.23	0.22	0.004
oth	others	0.14	0.14	.
SFA	Saturated FA	71.66	71.92	0.313
MUFA	Monounsaturated FA	25.09	24.80	0.266
PUFA	Polyunsaturated FA	3.25	3.29	0.102
n-3		0.448	0.445	0.023
n-6		2.15	2.21	0.069
*Nutritional Index*				
UFA/SFA		0.395	0.390	0.007
PUFA/SFA		0.046	0.046	0.002
n-6/n-3		4.840	4.985	0.161
AI	Atherogenic index	2.953	3.010	0.058
TI	Thrombogenic index	3.910	3.965	0.069
h/H	Hipocholesterolemic/hypercholesterolemic ratio	0.468	0.463	0.010

SEM, Standard Error of the Mean.

**Table 5 foods-12-04095-t005:** Mineral contents of mozzarella cheese from control (C) and Spirulina (S) groups.

Mineral (mg/100 g of Sample)		Group		SEM
*Abbreviation*	*Common Name*	C	S	
Ca	Calcium	248.093	248.228	25.388
Fe	Iron	0.323	0.285	0.037
K	Potassium	32.435	25.570	5.371
Mg	Magnesium	13.933	12.558	0.788
Mn	Manganese	19.880	16.365	4.724
Na	Sodium	93.263	101.718	10.848
P	Phosphorus	188.253	185.245	14.608
Zn	Zinc	1.773	1.615	0.224
Cu	Copper	0.143	0.095	0.020

SEM, Standard Error of the Mean.

**Table 6 foods-12-04095-t006:** Individual VOCs and class levels expressed as area unit ((AU) × 10^4^/g of product) in mozzarella cheese from control (C) and Spirulina (S) groups.

VOCs	RT	Match Factor	MZ	Group		SEM	*p*
				C	S		
* **Aldehyde** *				201.38	128.30	44.87	0.2934
Propanal, 2-methyl-	4.4	93.6	72	4.44	1.17	1.98	0.2872
Pentanal	12.1	95.2	58	9.59	2.58	2.23	0.0684
Hexanal	18.5	98.5	56	61.62	53.06	19.68	0.7690
Benzeneacetaldehyde	30.2	85.4	91	17.49	0.71	10.77	0.3130
Heptanal	23.5	96.5	70	5.27	6.61	2.18	0.6784
Benzaldehyde	26.7	98.7	106	44.87	59.03	21.52	0.6581
Butanal, 3-methyl-	8.9	97.2	58	44.37	2.30	18.28	0.1547
Butanal, 2-methyl-	9.5	94.0	58	13.73	2.84	8.11	0.3791
* **Ketone** *				451.35	239.79	169.09	0.4104
2,3-Pentanedione	12.5	90.9	100	3.94	4.78	1.59	0.7216
Cyclobutanone, 2,2,3-trimethyl-	14.7	92.1	55	3.33	1.50	1.10	0.2816
2-Heptanone	23.2	97.9	58	24.23	57.56	26.07	0.4008
2-Butanone	6.0	91.5	72	2.22	4.11	1.00	0.2296
Acetoin	14.9	96.2	45	406.57	158.01	155.26	0.3008
2-Pentanone	11.8	95.1	86	3.82	4.36	1.49	0.8083
Cyclohexanone	23.9	87.7	98	0.47	0.40	0.18	0.8010
2-Hydroxy-3-pentanone	20.3	90.4	45	3.93	2.99	1.09	0.5653
2-Octanone	27.4	90.6	58	0.19	0.48	0.16	0.2730
2-Nonanone	31.2	95.8	58	3.66	5.60	1.51	0.3988
* **Alcohol** *				498.54	303.11	232.63	0.5742
1-Propanol, 2-methyl-	8.4	96.2	43	14.25	6.73	8.70	0.5634
1-Pentanol	17.6	97.9	55	12.16	22.64	1.76	0.0057
1-Hexanol	22.6	98.1	56	15.31	39.64	12.49	0.2174
2-Heptanol	23.7	88.8	45	2.75	2.34	1.46	0.8489
Phenylethyl Alcohol	33.1	86.3	91	81.30	1.66	45.33	0.2702
Isopropyl Alcohol	10.8	86.1	45	10.58	32.60	16.28	0.3758
1-Hexanol, 4-methyl-	26.8	93.4	70	0.76	1.76	0.30	0.0563
2-Ethyl-1-hexanol	29.0	87.9	57	2.10	2.02	0.31	0.8595
Cyclobutanol	1.9	85.8	44	96.12	124.01	51.81	0.7166
1-Propanol	5.0	94.3	59	0.86	0.82	0.29	0.9331
1-Butanol, 3-methyl-	15.8	99.0	55	249.04	49.89	108.72	0.2428
1-Butanol, 2-methyl-	16.0	98.3	57	33.08	17.84	16.38	0.5351
1-Heptanol	26.8	89.8	55	0.59	1.17	0.22	0.1058
* **Ester** *				243.88	157.26	57.08	0.3245
Butanoic acid, ethyl ester	18.0	96.6	88	18.17	7.25	10.01	0.4699
Acetic acid, butyl ester	18.8	88.2	56	45.54	36.38	22.32	0.7815
1-Butanol, 3-methyl-, acetate	21.9	97.1	55	11.87	10.54	8.25	0.9129
Hexanoic acid, ethyl ester	27.0	96.1	88	15.86	4.17	11.11	0.4846
Sulphuric acid dibutyl ester	4.1	88.2	56	42.08	48.27	15.07	0.7813
CH_3_C(O)OCH(CH_3_)C(O)CH_3_ (*Acetoin acetate*)	24.1	90.5	87	0.31	0.52	0.30	0.6337
n-Caproic acid vinyl ester	27.0	87.3	99	10.96	5.03	6.35	0.5337
Propanoic acid, pentyl ester	30.7	85.9	70	0.43	0.42	0.10	0.9127
Acetic acid ethenyl ester	5.8	96.3	86	35.55	21.50	11.36	0.4156
Ethyl Acetate	6.2	98.1	70	39.18	11.96	21.20	0.3987
Propanoic acid, ethyl ester	12.5	97.7	57	17.04	7.96	7.80	0.4419
n-Propyl acetate	12.8	97.4	61	1.31	1.86	0.86	0.6696
Isobutyl acetate	16.6	93.8	73	2.31	0.42	0.51	0.0471
1-Butanol, 2-methyl-, acetate	22.0	89.0	70	2.93	0.87	1.63	0.4077
Acetic acid, 2-phenylethyl ester	36.9	90.2	104	3.71	0.15	0.17	0.0048
* **Aromatic hydrocarbons** *				308.86	244.99	17.46	0.0414
D-Limonene	27.7	98.2	93	28.32	25.51	4.65	0.6841
Styrene	22.5	92.1	104	5.86	5.28	0.79	0.6187
Toluene	15.6	99.0	91	253.91	195.22	16.29	0.0436
Ethylbenzene	20.8	98.6	91	6.71	6.15	0.85	0.6537
Benzene, 1,3-dimethyl-	21.2	99.2	91	14.05	12.84	1.60	0.6118
* **Hydrocarbons** *				426.30	330.39	73.77	0.3934
Hexane, 2,2-dimethyl-	8.7	97.1	57	16.22	15.32	2.38	0.7976
Dodecane	33.3	98.5	57	56.37	32.22	7.55	0.0644
Cyclopentane	3.8	92.3	55	2.24	2.43	0.49	0.8032
Heptane	9.5	97.4	100	7.28	10.65	4.79	0.6361
Undecane	29.8	97.3	85	6.47	3.98	1.26	0.2131
n-Hexane	4.5	98.8	57	80.41	101.17	25.07	0.5796
Heptane, 2,2-dimethyl-	25.9	91.2	57	6.21	3.16	1.95	0.3107
Undecane, 2,8-dimethyl-	29.6	87.6	71	0.50	1.63	0.64	0.2670
Pentane, 3,3-diethyl-	30.7	86.6	57	0.48	0.53	0.18	0.8647
Tridecane	36.5	98.3	57	31.39	18.24	4.34	0.0761
Heptane, 4-methyl-	14.4	96.6	70	5.36	5.10	3.19	0.9554
Cyclopentane, 1-ethyl-2-methyl-	16.1	89.2	55	106.15	7.85	65.74	0.3311
Octane	16.4	95.7	85	6.89	7.10	1.20	0.9030
Heptane, 2,3-dimethyl-	19.6	92.1	84	0.55	0.81	0.21	0.4062
Octane, 4-methyl-	19.9	97.3	85	3.29	3.45	1.99	0.9558
Decane	26.0	94.3	57	5.52	4.34	1.35	0.5606
Decane, 4-methyl-	26.4	91.9	71	2.19	2.38	1.25	0.9184
Nonane, 2,6-dimethyl-	26.6	91.0	71	2.26	2.66	1.38	0.8474
Decane, 2,4-dimethyl-	28.3	91.6	85	1.36	1.90	0.38	0.3632
3-Ethyl-3-methylheptane	28.6	90.0	71	0.41	0.62	0.28	0.6143
Undecane, 4,7-dimethyl-	30.0	89.3	71	1.87	2.72	0.62	0.3740
Pentane, 2-methyl-	3.7	91.5	71	2.77	2.29	0.27	0.2552
Pentane, 3-methyl-	4.1	96.2	57	80.10	100.40	24.93	0.5856
* **Halogenated hydrocarbons** *				141.86	32.90	81.34	0.3801
Trichloromethane	6.8	97.8	83	141.58	32.17	81.24	0.3777
Pentane, 3-bromo-	29.1	87.6	71	0.29	0.73	0.43	0.4955
* **Nitrogen compounds** *				10.50	8.41	1.74	0.4297
2-Propen-1-amine	25.9	91.2	57	6.21	3.29	1.94	0.3288
Diazene, dimethyl-	3.1	95.7	58	4.29	5.12	1.43	0.6931
* **Sulphur compounds** *				21.77	17.44	9.23	0.7515
Disulfide, dimethyl	14.5	96.2	94	18.89	15.26	8.44	0.7711
Dimethyl trisulfide	26.2	93.2	126	2.08	1.78	0.78	0.7956
Dimethyl sulfide	3.2	97.8	62	0.80	0.40	0.51	0.6018
* **Others** *				529.14	571.37	159.96	0.8581
Dimethyl ether	2.7	92.4	45	157.64	131.45	92.96	0.8487
Methylene chloride	3.7	93.7	84	3.82	3.54	0.31	0.5382
1,3-Dioxolane, 2,4,5-trimethyl-	13.1	91.6	101	2.59	3.60	0.63	0.2991
Dimethylphosphinic fluoride	10.9	85.4	81	2.12	3.27	0.37	0.0665
Bicyclo[3.1.1]hept-2-ene, 3,6,6-trimethyl-	23.6	93.0	93	11.40	11.13	1.17	0.8762
Methane, trimethoxy-	6.0	87.4	75	22.01	20.55	3.09	0.7500
1-Hexene	4.4	92.0	56	42.20	52.79	13.13	0.5891
2,4-Dimethyl-1-heptene	19.0	97.6	70	8.14	7.37	4.59	0.9091
Bicyclo[3.1.0]hex-2-ene, 2-methyl-5-(1-methylethyl)-	23.6	90.6	93	11.40	11.13	1.17	0.8762
Ethoxyacetylene	33.7	95.9	70	1.07	0.81	0.18	0.3384
Silanediol, dimethyl-	17.4	98.6	77	260.02	320.82	109.84	0.7090
tert-Butyl Hydroperoxide	19.9	93.1	59	6.04	4.40	1.90	0.5656
Butanal, 3,3-dimethyl-2-oxo-, hemihydrate	34.0	91.5	85	0.69	0.51	0.12	0.3245

SEM, Standard Error of the Mean.

**Table 7 foods-12-04095-t007:** Values (mean ± S.E.) of CIE L*, a*, and b* of the external and internal surfaces of mozzarella cheese from control (C) and Spirulina (S) groups.

	External Surface			Internal Surface		
	Group		*p*	Group		*p*
	S	C		S	C	
L*	93.43 ± 0.02	92.91 ± 0.04	<0.0001	93.13 ± 0.06	92.80 ± 0.15	0.0410
a*	−2.26 ± 0.03	−2.13 ± 0.03	<0.0001	−3.42 ± 0.03	−3.13 ± 0.02	<0.0001
b*	0.62 ± 0.07	0.41 ± 0.01	0.004	4.14 ± 0.05	3.39 ± 0.05	<0.0001

**Table 8 foods-12-04095-t008:** List of attributes assessed by panellists during the quantitative descriptive analysis (QDA) of mozzarella cheeses from control (C) and Spirulina (S) groups.

Sensory Attribute	Group		SEM	*p*
	C	S		
*Appearance*	75.70	74.28	3.33	0.7660
Colour uniformity	39.45	40.62	2.61	0.7540
Surface uniformity	53.21	40.13	2.67	0.0030
Inner colour	56.14	66.27	2.70	0.0160
Eye number	23.93	27.86	3.43	0.4290
Uniformity after cutting	58.55	58.53	3.40	0.9970
Buttermilk release	44.15	67.32	3.91	0.0006
*Odour*				
Milk	44.80	50.51	2.85	0.1730
Butter	33.36	42.89	2.22	0.0070
Yoghurt	20.77	24.96	1.68	0.0940
*Taste*				
Saltiness	25.78	24.72	2.90	0.7980
Sweetness	12.87	17.72	1.23	0.0125
Bitterness	12.04	16.87	1.55	0.0415
Sourness	15.57	18.79	2.27	0.3290
*Flavour*				
Fruity	12.82	15.17	2.05	0.4270
Whey	43.66	53.77	2.77	0.0190
*Texture*				
Tenderness	51.47	59.02	2.85	0.0710
Oiliness	28.12	44.13	3.28	0.0030
Moisture	55.76	78.71	2.12	<0.0001
Graininess	33.09	20.59	2.36	0.0010
Cohesiveness	60.57	27.79	3.57	<0.0001
Screechy	50.64	31.70	3.97	0.0030
Shear consistency	66.83	48.07	1.78	<0.0001

SEM, Standard Error of the Mean.

**Table 9 foods-12-04095-t009:** Rating attributed by the consumers (%) to mozzarella cheese from control (C) and Spirulina (S) groups in relation to absence (perceived) or presence (actual) of information and expectation (expected).

Hedonic Scale	Perceived		Expected		Actual	
	C	S	C	S	C	S
Dislike extremely—1	0	0	0	0	0	0
Dislike very much—2	0	3	0	0	2	0
Dislike moderately—3	1	5	0	2	0	0
Dislike slightly—4	4	8	2	2	0	5
Neither like nor dislike—5	4	6	5	6	12	8
Like slightly—6	25	20	6	19	18	11
Like moderately—7	38	26	33	29	34	23
Like very much—8	19	29	43	35	32	34
Like extremely—9	7	5	11	8	2	19

**Table 10 foods-12-04095-t010:** Mean rating (given by the consumers) during the three hedonic tests of mozzarella cheese from control (C) and Spirulina (S) groups.

Acceptability	Group		*p*
	C	S	
Perceived (P)	6.81	6.43	0.0903
Expected (E)	7.44	7.07	0.1034
Actual (A)	6.85	7.29	0.0508
P-E	−0.63 *	−0.64 *	
A-P	0.04	0.86 **	
A-E	−0.59	0.22	

* *p* < 0.05; ** *p* < 0.01.

## Data Availability

The datasets of the present study are available from the corresponding authors upon reasonable request.

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
