# Peer review of "Use of Cyanobacterium Spirulina (Arthrospira platensis) in Buffalo Feeding: Effect on Mozzarella Cheese Quality"

_foods, 2023, doi:10.3390/foods12224095_

Round 1

Reviewer 1 Report

Comments and Suggestions for Authors

The revision is attached as a separate document

Author Response

Reviewer 1

We sincerely thank the reviewer for his competent work and valuable comments, which helped clarify our manuscript. The line number refers to the new version with the tracked changes. We wanted to warn you that some part of the text has been extensively changed to minimize the repetition rate. Accordingly, these changes are also visible in track change mode.

General comments

  • REV. The experiment group received 50 g or 100 g spirulina additive (during 120 days in total). Please explain what was the reason the use of such a supplementation period. Also the amount of additive seems too low to achieve effect in milk composition? Please explain.
  1. Thank you for your comment. The dose of Spirulina added to the diet was chosen considering the results of other studies (see, in particular, Simkus et al., 2008; Christaki et al., 2012) and our preliminary palatability tests on buffalo cows. The choice of a 120-day trial period was based on the need to test the effect of Spirulina in the most important production phases of a buffalo's lactation (i.e., first lactation, peak lactation and the stationary phase), which in total lasts approximately 270 days.
  • The main weakness refers to sensory performing a sensory assessment. Why sensory estimation was not conducted by experts ? Accordingly recommendations only experts can participate in quantitative descriptive analysis. Have you received consents and agreement for studies with humans, for this estimation (participants were recruited)?
  1. Regarding the suggestion that only professionals should be used to carry out QDA, we recognize the feasibility of using a trained group for this task. As stated in the M&M (see lines 280-300), we enlisted typical mozzarella consumers and assessed their ability to discriminate the basic tastes. This preliminary procedure and the subsequent training phases of the panelists until the judgement is reached are like those detailed in our previous works on mozzarella (e.g., Uzun et al., 2018; Serrapica et al., 2022) and other stretched curd cheeses (e.g., Esposito et al., 2014; Braghieri et al., 2018; Serrapica et al., 2019 among others). Regarding consent and agreement, we received them (see lines 304-305). Thanks.
  • In consumer estimation participated not sufficient number of member. Accordingly recommendation should be 100 or more. Please explain why you use 68 consumers in the study?
  1. The lower-than-normal number of consumer trial participants (68 vs 100) was due to the COVID-19 pandemic period, which made participant recruitment challenging. We added this information also in paragraph 2.3.2 (lines 256-257). Thanks.
  • The results of quantitative descriptive analysis are not poorly presented. Figure 1 is not presented the scoring intensity of sensory attributes and significant differences in two samples of cheeses (experimental - S and control). Please prepare Table. Did you find some differences in aroma or taste (flavour attributes). Please supply. In such described results is difficult to find. Also description of QDA results should be presented in the same order as attributes were presented in Table 2. Please prepare suggested Table and revise text accordingly.
  1. Your suggestion has been accepted and, accordingly, table has been added to the text (named Table 8, line 489). Differences in sensory attributes are reported in the main text at paragraph 3.2 (lines 493-509). Thanks.
  • Also results of consumer estimation are also not properly presented in Table 8. The results clearly deviate from normality. Instead of mean values Authors should present number of responses for each rate: (distribution of individual preferences of two samples of cheeses). Please prepare appropriate Table or Figure and revise text.
  1. According to the reviewer's suggestion, a new table was added to the text (Table 9). When discussing the results of the QDA, we used mean values and analyzed the data using ANOVA according to Stone and Sidel (2004). (Sensory evaluation practice-3rd ed, Elsevier Academic Press, Ch3, p.89) "Parametric statistical analysis, such as AOV, of data from the nine-point hedonic scale can provide useful information about differences between products, and it should not be assumed that data from this scale violate the normality assumption". Thanks.
  • Regarding results of volatile compounds part 3.1. I suggest presenting all results in main text not in supplementary material. Also please arrange compounds into chemical groups. Please verify if there some significant differences in amount of compounds responsible for aroma attribute between groups and revise text.
  1. As for reviewer suggestion, volatile compounds table are added to main text (Table 6), compounds arranged for chemical groups (Table 6), and main text was changed accordingly.
  • Regarding part 3.1: why you did not present sum of EAA and NEAA (maybe you find some interesting differences).
  1. According to the reviewer suggestion, the sum of both EAA and NEAA were reported in Table 3. Thanks.
  • Generally, you find some significant differences in intensity of sensory attributes you should find the reason and try to explain why? Please supply.
  1. In accordance with what has been suggested, we have tried to summarize the issue in lines 504-507. Based on the obtained results, it is not easy to identify the main cause of the observed differences and, therefore, we are planning to investigate these aspects further in the near future.
  • Part 4: In my opinion there is a lack of conclusion please prepare
  1. Conclusion has been totally rearranged according to the reviewer suggestion.

Other specific comments

  • Line 57: Please describe crucial results of cited Authors [12-15] (one/two sentences).
  1. Since the cited papers are literature reviews, we have opted to specify this in the introduction omitting further details because, as later detailed (lines 383-388), the reported findings are frequently conflicting.

REV. Lines 74-78: please consider to reformulate the aim. It is not properly regarding English grammar. Also the sentence in Lines 76-78 is not completed please supply. I advise not using acronym (WTP) in the aim (please consider)

  1. Thank you for your suggestion. We revised the text following the reviewer's suggestions (lines 73-79).
  • Line 76: Should be “freeze-dried”, please correct.
  1. Thanks for the remark. We have changed (line 75).
  • Part 2.2.3. please supply information about quantitative analysis and used standards.
  1. The requested information was added to the FA analysis description (2.2.3).
  • Line please explain used quantitative method for Volatile Compounds (presented in Line 188).
  1. Thanks for the remark. The change has been made (line 200).
  • The names of some attributes and definitions presented in Table 2 should be corrected with regard in English. Please verify.
  1. We have checked the attributes listed in Table 2 for reviewer suggestions. Thanks.

Reviewer 2 Report

Comments and Suggestions for Authors

The paper entitled “Use of cyanobacterium Spirulina (Arthrospira platensis) in buffalo feeding: effect on mozzarella cheese quality” is well written. The experimental design is correct and the results are interesting.

The discussion of the sensory analysis results can be improved. The authors detect differences between S and C cheeses. This means that the addition of spirulina affects the sensory characteristics of cheese. The authors describes possible causes that could explain this sensory differences. However, in this work no differences in composition, free fatty acids, free amino acids, etc. are detected. How do they explain the sensory differences?

The information give to the consumers about S cheese for the Expected liking "Dietary supplementation with Spirulina (a natural product arising from the freeze-drying of microalgae) makes it possible to obtain a mozzarella with antioxidant, anti-cholesterol, and immune-stimulating properties" not corresponded to the real cheeses, because S cheese doesn't had differences in nutritional index (compared to control cheeses). The information obtained about Expected liking and Actual liking is equally interesting, but it should be clarified in the paper that, if the concentration of spirulina is increased, in order to obtain better nutritional index, the sensory characteristics of cheeses will be different. So the tasters will not rate the same Actual liking.

In the conclusion, it will be better if authors include the concentration of spirulina used.

In line 407: I think there is a mistake. Table S1 was Table S2

In line 419: I think there is a mistake. Table 4 was Table 6.

Author Response

Reviewer 2

We thank the reviewers for the time spent in reviewing our manuscript. The line number refers to the new version with the tracked changes. We wanted to warn you that some part of the text has been extensively changed to minimize the repetition rate. Accordingly, these changes are also visible in track change mode. In addition, to comply with the request of reviewer 1, the VOCs table presented as supplementary material has been added to the main test. For the same reason, the results of the sensory analysis have been presented in a table rather than a graph and a new table, Table 9, has been added.

  • The paper entitled “Use of cyanobacterium Spirulina (Arthrospira platensis) in buffalo feeding: effect on mozzarella cheese quality” is well written. The experimental design is correct and the results are interesting.
  1. Thank for the kind feedback and for the competent work which has helped us to really improve our manuscript.
  • The discussion of the sensory analysis results can be improved. The authors detect differences between S and C cheeses. This means that the addition of spirulina affects the sensory characteristics of cheese. The authors describes possible causes that could explain this sensory differences. However, in this work no differences in composition, free fatty acids, free amino acids, etc. are detected. How do they explain the sensory differences?
  1. Thanks for the remarks. We rearranged the text to put more emphasis on the possible causes of influence.
  • The information give to the consumers about S cheese for the Expected liking "Dietary supplementation with Spirulina (a natural product arising from the freeze-drying of microalgae) makes it possible to obtain a mozzarella with antioxidant, anti-cholesterol, and immune-stimulating properties" not corresponded to the real cheeses, because S cheese doesn't had differences in nutritional index (compared to control cheeses). The information obtained about Expected liking and Actual liking is equally interesting, but it should be clarified in the paper that, if the concentration of spirulina is increased, in order to obtain better nutritional index, the sensory characteristics of cheeses will be different. So the tasters will not rate the same Actual liking.
  1. Thanks for the remarks. The information provided to the panellists concerning the potential beneficial effects of Spirulina were obtained from literature and formed the hypothesis investigated in our research. According to the reviewer suggestion, we have rearranged the discussions and conclusion.
  • In the conclusion, it will be better if authors include the concentration of spirulina used.
  1. Thanks for the suggestion. The conclusions have been updated (L 594-605).
  • In line 407: I think there is a mistake. Table S1 was Table S2.
  1. We apologize for the inconvenience. The text has been changed. In the current form of the manuscript, Table S2 has been integrated into the main text as Table 6, according to Rev.1 suggestion. We thank you for the careful report.
  • In line 419: I think there is a mistake. Table 4 was Table 6.
  1. We have changed according to the reviewer's suggestion (L 409). Thanks.

Reviewer 3 Report

Comments and Suggestions for Authors

In the manuscript entitled "Use of cyanobacterium Spirulina (Arthrospira platensis) in buffalo feeding: effect on mozzarella cheese quality", the authors stated the effects on the chemical composition and sensory quality of PDO buffalo mozzarella cheese produced by supplementing dairy buffalo diet with freeze-dried Spirulina platensis biomass. Also, they investigated consumers' acceptability and willingness to pay (WTP). The manuscript is engaging, and the results are presented in detail. However, I have some minor questions and comments.

Introduction

Please correct a typo in L75: freeze-died.

Materials and Methods

L24: Please include a brief description of the method of manufacturing mozzarella cheese.

L137-138: Were the cheese samples minced in whole? What instrument was used to mince the cheese samples?

L138: "moisture" would be a more correct term than "humidity".

L138-139: What cheese weights were analyzed?

L164-169: What weights of milk and cheese were analyzed? Under what conditions were the samples mineralized? What mineral compounds were determined? Please complete.

L310: In what software were the results statistically analyzed?

Results

L388-391: Perhaps it would be useful to briefly describe the results of the analysis of mineral levels in milk?

Conclusions

I suggest briefly describing how the sensory characteristic of mozzarella cheese and information communication to consumers impact their liking and willingness to pay.

References

The cited publication references in half refer to articles from the last five years, but citations to older publications are reasonable.

Author Response

Reviewer 3

We thank the reviewers for the time spent in reviewing our manuscript. Line number refers to the new version with the tracked changes. We wanted to warn you that some part of the text has been extensively changed to minimize the repetition rate. Accordingly, these changes are also visible in track change mode. In addition, to comply with the request of reviewer 1, the VOCs table presented as supplementary material has been added to the main test. For the same reason, the results of the sensory analysis have been presented in a table rather than a graph and a new table, Table 9, has been added.

  • The manuscript is engaging, and the results are presented in detail.

- We thank the reviewer for the positive feedback, and the competent work which has helped us to clarify our manuscript.

  • Please correct a typo in L75: freeze-died.

- Thank you for the useful suggestion. We have corrected the mistake at L75.

  • L124: Please include a brief description of the method of manufacturing mozzarella cheese.

- Thanks for the remark. According to the reviewer's suggestion, a brief description of the manufacturing mozzarella cheese procedure has been added to the text (L124-129), as follows: ”In brief, the raw milk was heated to 37°C and natural starter cultures and liquid calf rennet were added. The curd was then broken into small particles (2-3 cm) and left under whey until the pH reached 4.85. At this pH, the curd was manually stretched in water kept at 90-95°C. Finally, 50 g of mozzarella (spherical shape) were cut mechanically, cooled in water, and left in brine (10% NaCl)".

  • L137-138: Were the cheese samples minced in whole? What instrument was used to mince the cheese samples?

- Thanks for the remark. To dispel concerns about the sample mincing procedure, the following statement has been added (see L 142-143): "Samples were cut in slices and then minced by using a blender". In addition, the device used for mincing samples has been specified at L 143 (LB20ES, Waring Commercial Blender, New Hartford, CT, USA).

  • L138-139: What cheese weights were analyzed?

- As suggested, we have updated the weight of the analyzed samples to L144. Thanks.

  • L164-169: What weights of milk and cheese were analyzed? Under what conditions were the samples mineralized? What mineral compounds were determined? Please complete.

- Thanks for the useful suggestions. The required information has been added to the text at L171-175.

  • L310: In what software were the results statistically analyzed?

- We statistically analyzed the results using SAS software. This information has been added to the text, at L 338-339. Thanks.

  • L388-391: Perhaps it would be useful to briefly describe the results of the analysis of mineral levels in milk?

- We apologize for writing about the mineral analysis of milk and mozzarella cheese, but in this article, as is evident, only the mozzarella cheese is mentioned and not the composition of the milk. We have rectified the text accordingly (L 170-180). We apologize for the unfortunate inconvenience. We thank you for the careful report.

  • I suggest briefly describing how the sensory characteristic of mozzarella cheese and information communication to consumers impact their liking and willingness to pay.

- Thanks for the useful remark. We approached the reviewers’ suggestions throughout the discussions.

  • The cited publication references in half refer to articles from the last five years, but citations to older publications are reasonable.

- Thank for the kind feedback.

Round 2

Reviewer 1 Report

Comments and Suggestions for Authors

The manuscript was sufficiently revised by the Authors. I also appreciate responses on all my comments. Thank you,